# Vitamin K2 Modulates Mitochondrial Dysfunction Induced by 6-Hydroxydopamine in SH-SY5Y Cells via Mitochondrial Quality-Control Loop

**DOI:** 10.3390/nu14071504

**Published:** 2022-04-04

**Authors:** Hengfang Tang, Zhiming Zheng, Han Wang, Li Wang, Genhai Zhao, Peng Wang

**Affiliations:** 1Institute of Intelligent Machines, Hefei Institutes of Physical Science, Chinese Academy of Sciences, Hefei 230031, China; hengfangt@163.com (H.T.); wh_yingwang@163.com (H.W.); liwang@ipp.ac.cn (L.W.); zhgh327@ipp.ac.cn (G.Z.); 2Science Island Branch of Graduate, University of Science and Technology of China, Hefei 230026, China; 3Anhui Key Laboratory of Environmental Toxicology and Pollution Control Technology, Hefei Institutes of Physical Science, Chinese Academy of Sciences, Hefei 230031, China; 4CAS (Hefei) Institute of Technology Innovation Co., Ltd., Hefei 230088, China

**Keywords:** vitamin K2, apoptosis, ROS, mitochondrial dysfunction, mitophagy, mitochondrial biogenesis, mitochondrial quality-control loop, PINK1/Parkin signaling pathway

## Abstract

Vitamin K2, a natural fat-soluble vitamin, is a potent neuroprotective molecule, owing to its antioxidant effect, but its mechanism has not been fully elucidated. Therefore, we stimulated SH-SY5Y cells with 6-hydroxydopamine (6-OHDA) in a proper dose-dependent manner, followed by a treatment of vitamin K2. In the presence of 6-OHDA, cell viability was reduced, the mitochondrial membrane potential was decreased, and the accumulation of reactive oxygen species (ROS) was increased. Moreover, the treatment of 6-OHDA promoted mitochondria-mediated apoptosis and abnormal mitochondrial fission and fusion. However, vitamin K2 significantly suppressed 6-OHDA-induced changes. Vitamin K2 played a significant part in apoptosis by upregulating and downregulating Bcl-2 and Bax protein expressions, respectively, which inhibited mitochondrial depolarization, and ROS accumulation to maintain mitochondrial structure and functional stabilities. Additionally, vitamin K2 significantly inhibited the 6-OHDA-induced downregulation of the MFN1/2 level and upregulation of the DRP1 level, respectively, and this enabled cells to maintain the dynamic balance of mitochondrial fusion and fission. Furthermore, vitamin K2 treatments downregulated the expression level of p62 and upregulated the expression level of LC3A in 6-OHDA-treated cells via the PINK1/Parkin signaling pathway, thereby promoting mitophagy. Moreover, it induced mitochondrial biogenesis in 6-OHDA damaged cells by promoting the expression of PGC-1α, NRF1, and TFAM. These indicated that vitamin K2 can release mitochondrial damage, and that this effect is related to the participation of vitamin K2 in the regulation of the mitochondrial quality-control loop, through the maintenance of the mitochondrial quality-control system, and repair mitochondrial dysfunction, thereby alleviating neuronal cell death mediated by mitochondrial damage.

## 1. Introduction

Neurodegenerative disease collectively refers to a group of neurological diseases, including the progressive degeneration or death of neurones [1]. Parkinson’s disease (PD) is the second most common neurodegenerative disorder, and it is characterized by a progressive loss of dopaminergic neurones in the substantia nigra pars compacta (SNpc) and the presence of Lewy bodies in the remaining dopamine neurones [2]. The pathogenesis of PD is complex, and the exact etiology and natural process have not been fully defined. However, previous studies indicated that mitochondrial dysfunction, autophagy failure, and neuroinflammation play a role [3,4,5], and that mitochondrial dysfunction can exacerbate neuronal degeneration [6]. Many proteins encoded by PD-related genes identified by genome-wide association studies (GWASs) and Mendelian inheritance patterns have been proved to play direct or indirect roles in mitochondrial homeostasis or mitochondrial autophagy [7].

Mitochondria are the energy metabolism centers of cells that produce ATP through the electron transport chain to provide cells with the energy needed for survival, whereas reactive oxygen species (ROS) are an inevitable product of mitochondrial respiration. When cells are damaged, the function of the respiratory chain complex is impaired, ATP synthesis is reduced, and the clearance of oxygen free radicals and ROS is obstructed, resulting in an imbalance in cellular oxidative stress. Oxidative-stress imbalance produces a large number of ROS that can damage mitochondrial proteins, resulting in impaired membrane integrity, increased permeability, and decreased membrane potential, and leads to mitochondrial dysfunction and cell death [8,9]. Mitophagy specifically targets and degrades damaged mitochondria, thereby blocking the mitochondrial apoptosis cascade. Among them, the degradation products of amino acids, fatty acids, and other small-molecule compounds can be recycled by the body, providing a material basis for mitochondrial production [10]. Studies have shown that there is a close relationship between mitochondrial fission and the mechanism of mitophagy, and that mitochondrial fission is of great significance in the occurrence of mitophagy [11]. Under the control of a series of DRPs, damaged mitochondria are divided into two parts, the damaged part is degraded by mitophagy, and the resulting degraded products are reused to generate new mitochondria [12]. Pink1-dependent Parkin activation is believed to be the main pathway of mitophagy and is critical for mitochondrial quality control in many models; and an abnormal mechanism in this process can result in the accumulation of ROS in damaged mitochondria, leading to cellular oxidative stress [13]. Therefore, proper regulation of the PINK1/Parkin signaling pathway is an effective measure to block the expansion of mitochondrial damage. Based on previous studies, we determined that an imbalance in mitochondrial oxidative stress can lead to mitochondrial structural dysfunction and depletion of cellular energy reserves. Mitochondrial dysfunction may play a significant role in the development of PD, and neurodegeneration can be reduced by protecting mitochondria.

Vitamin K2, a natural fat-soluble vitamin, is represented by MK-n, according to the number of structural units of its side chain, isoprene. Currently, there are 14 known molecules in nature, including MK-1 and MK-14 [14]. Vitamin K2 plays a significant part in promoting coagulation and cardiovascular and bone metabolism [15]. In recent years, vitamin K2 has become an important bioactive compound, owing to its antioxidant action [16]. Accumulating evidence suggests that vitamin K2 inhibits inflammatory responses and repairs mitochondrial damage induced by oxidative stress in vitro. Studies have found that, when astrocytes are exposed to hypoxia, MK-7 pretreatment not only reduces neuroinflammation but also increases ATP production and inhibits ROS production [17]. In another research study, MK-4 rescued severe mitochondrial defects caused by mutations in the PINK1 gene [18]. In addition, vitamin K2 significantly inhibits the generation of ROS and the activation of p38 and caspase in microglia induced by rotenone [19].

However, the mechanism through which vitamin K2 targets mitochondrial dysfunction as an antioxidant remains unclear. To deepen the research in this direction, we investigated the signaling of vitamin K2 in the regulation of neuronal cell death due to mitochondrial dysfunction in 6-OHDA-treated SH-SY5Y cells. SH-SY5Y cells are similar to neurones and are a widely used neurodegenerative diseases model. Therefore, SH-SY5Y cells were used in this research [20]. To reveal the molecular mechanism, SH-SY5Y cells were treated with 6-OHDA, an effective dopamine-induced neuronal degeneration nerve agent that can induce mitochondrial damage, chronic oxidative stress, and cell death. It is widely used in the construction of cellular and animal PD models [21]. Our study suggests that vitamin K2 promotes mitochondrial fission and autophagy by activating the PINK1/Parink signaling pathway, thereby blocking the expansion of mitochondrial damage, promoting mitochondrial fusion and biogenesis, regulating the mitochondrial quality-control loop, and ultimately improving oxidative-stress-induced mitochondrial dysfunction.

## 2. Materials and Methods

### 2.1. Cell Culture

SH-SY5Y cells (Procell CL-0208) were kindly provided by *Procell Life Science & Technology Co., Ltd.* (Wuhan, China). Cells were seeded in Dulbecco’s modified Eagle medium/nutrient mixture F12 (DMEM/F12) (Biological Industries, Kibbutz Beit Haemek, Israel) containing 15% fetal bovine serum (FBS) (ExCell Bio, Shanghai, China) and antibiotics (100 U/mL penicillin G, 100 μg/mL streptomycin) (Beyotime, Shanghai, China) and maintained at 37 °C and 5% CO_2_ in a humidified incubator. For experimental assays, cells were trypsinized by using 0.2% trypsin (Beyotime).

### 2.2. Drug Treatment

To verify the mitigating effect of vitamin K2 (MK-7, extracted from *Bacillus subtilis (natto)* in our laboratory) on neuronal toxicity induced by 6-OHDA (MCE, Shanghai, China), we administered different concentrations of vitamin K2 (10, 20, 30, and 40 µM) and 6-OHDA (100, 200, 300, 400, and 500 µM) in a time-dependent dose-response manner. We determined that 300 µM for 2 h was the maximum dose for 6-OHDA, and vitamin K2 had the best inhibitory effect on 6-OHDA toxicity at a concentration of 30 µM. The following treatment groups were used: (1) control, (2) 6-OHDA, (3) vitamin K2, and (4) 6-OHDA + vitamin K2. For each experiment, when the cell density reached 80%, the cells were treated with 300 µM 6-OHDA for 2 h, followed by post-treatment with vitamin K2 (30 µM) for 6 h. All experiments were performed in triplicate or quadruplet, and all data presented were blank-corrected.

### 2.3. Cell Viability Assay

Cell viability was measured by using the Cell Counting Kit-8 (CCK-8) assay (Biosharp, Hefei, China). Briefly, after the treatment, the medium was aspirated, and 20 μL CCK-8 and 180 μL fresh medium were added to each well and incubated for 2 h at 37 °C in the dark. The absorbance was measured at 450 nm, using a microplate reader (Molecular Devices, Sunnyvale, CA, USA).

### 2.4. Reactive Oxygen Species (ROS) Assessment

The fluorescence probe 2′,7′-dichlorofluorescin diacetate (DCFH-DA, Sigma-Aldrich, Saint Louis, MO, USA) was used to measure the amount of intracellular ROS in cells. The cells were seeded in 12-well plates at a density of 5 × 10^5^ cells/well. When cells were treated with 6-OHDA and VK2, respectively, 5 µM DCFH-DA was added to each well and incubated for 30 min, at 37 °C, in the dark. The cells were then washed with PBS buffer three times and analyzed by using flow cytometry (EXFLOW-206, DAKEWE, Beijing, China). All images were obtained by using a fluorescence microscope (CellInsight CX5; Thermo Fisher Scientific, Waltham, MA, USA). The mean fluorescence intensity from three different fields of view and from three independent experiments was quantified by using ImageJ software.

### 2.5. Measurement of Apoptosis

Apoptosis was measured by using a propidium iodide (PI) probe (Beyotime). Briefly, the cells were seeded in 12-well plates at a density of 5 × 10^5^ cells/well. The cells were treated with 6-OHDA and VK2, respectively, added with PI, and incubated at 22 ± 3 °C for 20 min in the dark. The cells were then photographed by a Thermo CX5 HCS fluorescence microscope or analyzed by using flow cytometry.

### 2.6. Detection of Mitochondrial Membrane Potential

The JC-1 probe (Beyotime) was used to measure the mitochondrial membrane potential. Briefly, the cells were seeded at a density of 5 × 10^5^ cells/well in 12-well plates. The cells were then treated with 6-OHDA and VK2, respectively, an addition of 1 mg/mL JC-1 was made to each well, and they were incubated at 37 °C for 30 min in the dark. The cells were then washed with PBS buffer three times and photographed with a Thermo CX5 HCS fluorescence microscope. The monomer fluorescence (green) of JC-1 was observed at emission (Em) = 521 nm, with excitation (Ex) = 485 nm, and aggregated fluorescence (red) of JC-1 was detected at Em = 607 nm with Ex = 560 nm.

### 2.7. Real-Time Fluorescent Quantitative PCR Analysis

The cells were seeded in 6-well plates at a density of 4 × 10^5^ cells/well and washed twice with PBS after the treatment. The total RNA was extracted by using the RNA-easy Isolation Reagent (Vazyme, Nanjing, China). A HiScript II 1st Strand cDNA Synthesis Kit (Vazyme) was used to synthesize the cDNA. The sequences of the forward and reverse primers are listed as follows: Bax Fw primer: 5′-GGCCAATTGGAGATGAACTG-3′, Rev primer: 5′-GTCACTGTCTGCCATGTGGG-3′; Bcl-2 Fw primer: 5′-CCTGTGGATGACTGAGTACC-3′, Rev primer: 5′-GAGACAGCCAGGAGAAATCA-3′; MFN1 Fw primer: 5′-ATGGCAGAAACGGTATCTCC-3′, Rev primer: 5′-TTAGGATTCTCCACTGCTCG-3′; MFN2 Fw primer: 5′-TCCTTGAAGACACCCACAGGAA-3′, Rev primer: 5′-AGTTGGTTCACAGTCTTGACACTCT-3′; DRP1 Fw primer: 5′-ATGGAGGCGCTGATCCCGGTCATCA-3′, Rev primer: 5′-TCACCAAAGATGAGTCTCTCGGATTT-3′; P62 Fw primer: 5′-ATGAGGACGGGGACTTGGTTGCCTTTT-3′, Rev primer: 5′-TCACAACGGCGGGGGATGCTTTGAATACT-3′; LC3A Fw primer: 5′-ATGCCCTCAGACCGGCCTTTCAA-3′, Rev primer: 5′-TCAGAAGCCGAAGGTTTCCTGGGAGGCG-3′; PGC-1α Fw primer: 5′-GCAGTCGCAACATGCTCAAG-3′, Rev primer: 5′-GGGAACCCTTGGGGTCATTT-3′; NRF1 Fw primer: 5′-ATGGAGGAGCACGGAGTGACCCAAA-3′, Rev primer: 5′-CTACTGTTCCAAGGTCACCACCTCCACA-3′; TFAM Fw primer: 5′-ATGGCGCTGTTCCGGGGAATGTGGA-3′, Rev primer: 5′-TTAATGCTCAGAGATGTCTCCGGATCGTTT-3′; PINK1 Fw primer: 5′-ATGGCGGTGCGACAGGCACT-3′, Rev primer: 5′-TCATGGGGCTGCCCTCCAGGAAGAGA-3′; Parkin Fw primer: 5′-ATGATAGTGTTTGTCAGGTTCAACTCCAGC-3′, Rev primer: 5′-CTACACGTCGAACCAGTGGTCCCCCATGCA-3′; GAPDH Fw primer: 5′-CAATGTGTCCGTCGTGGATCTGA-3′, Rev primer: 5′-AAGGTGGAAGAGTGGGAGTTGCT-3′.

The RT-qPCR reaction system included 4 ng cDNA, 0.2 µM primer, 10 μL ChamQ Universal SYBR qPCR Master Mix (Vazyme), and ddH_2_O added to 20 μL. The PCR amplification was carried out using Rotor Gene (Cobas z 480; Roche, Switzerland) under the following conditions: 40 cycles of 30 s at 95 °C, 10 s at 95 °C, and 30 s at 60 °C. Relative mRNA expression was determined by using the 2^−△△Ct^ method and was normalized to GAPDH expression.

### 2.8. Western Blotting

After the treatment of cells with 6-OHDA and vitamin K2, respectively, cells were lysed with a lysis buffer (Beyotime) and centrifuged at 4 °C for 10 min at 12,000 rpm. A BCA assay kit (Sangon Biotech, Shanghai, China) was used to measure the concentration of total protein. For each group, 20 μg protein total proteins were separated by using 12% SDS–PAGE and transferred to a polyvinylidene fluoride (PVDF) membrane. After blocked with protein-free rapid blocking buffer (Epizyme Biotech, Shanghai, China) at 22 ± 3 °C for 20 min, the membranes were incubated with primary antibodies, including PGC-1α, PINK1, MFN1, MFN2, Nrf2, Bax, TFAM (Bimake, Houston, TX, USA), Bcl-2 (Cell Singling Technology, Boston, MA, USA) and β-actin (Proteintech, Wuhan, China), respectively, at appropriate dilutions, overnight, at 4 °C. After washing with TBST for 30 min, the membranes were incubated with anti-mouse or anti-rabbit HRP-conjugated secondary antibodies (Proteintech) at 22 ± 3 °C for 1 h. Finally, the PVDF membranes were then covered with a hypersensitive enhanced chemiluminescence (ECL) solution (Vazyme) and visualized by using a luminescent image analyzer (UVITEC, Mini HD9, Cambridge, UK).

### 2.9. Statistical Analyses

All data are presented as mean ± SD and were compared with the respective control groups. Statistical significance was assessed by one-way ANOVA, using GraphPad Prism 9 software, and differences with *p* ≤ 0.5 were designated as significant in all cases.

## 3. Results

### 3.1. Vitamin K2 Protects SH-SY5Y Cells from 6-OHDA-Induced Reduction in Cell Viability

First, the effect of 6-OHDA on SH-SY5Y cell viability was detected by using the CCK-8 method. Then the cells were treated with 6-OHDA (0–500 µM) for 2 h in a dose-dependent manner. Compared with the control group, the number of viable cells was significantly reduced and the lethality of cells reached 50% at a concentration of 300 µM. Therefore, the cells treated with 6-OHDA at this concentration were selected for subsequent experiments. Subsequently, the damaged cells recovered to a certain extent after treatment with vitamin K2 (0–40 µM) for 6 h, indicating that vitamin K2 inhibited the toxic effect of 6-OHDA, and that the inhibitory effect was the best at a concentration of 30 µM. Therefore, the concentration of vitamin K2 used in subsequent experiments was 30 µM, as shown in Figure 1.

### 3.2. Vitamin K2 Has a Protective Effect on 6-OHDA-Mediated Apoptosis of SH-SY5Y Cells

The anti-apoptotic effect of vitamin K2 on SH-SY5Y cells was investigated by using PI staining. As shown in Figure 2A, when 6-OHDA was treated, the apoptosis rate increased from 4.28% in the control group to 20.44%. However, after vitamin K2 post-treatment, the apoptosis rate was significantly reduced to 11.44%. In addition, the fluorescence microscopy results showed that the intracellular PI fluorescence signal was significantly increased after 6-OHDA treatment, indicating an increase in apoptotic cells. However, post-treatment with vitamin K2 reduced the PI fluorescence signal. (Figure 2B). In addition, through the study of vitamin K2 on the expression of apoptosis signaling proteins Bax and Bcl-2 in SH-SY5Y cells, it was found that vitamin K2 could significantly inhibit the ratio of mRNA and protein expression level on pro-apoptotic protein (Bax) and anti-apoptotic protein (Bcl-2) (Figure 2C,D). These results indicate that vitamin K2 could inhibit apoptosis induced by 6-OHDA and alleviate cell damage.

### 3.3. Vitamin K2 Relieves Oxidative Stress Caused by 6-OHDA

The potent nerve agent 6-OHDA induces oxidative stress, produces substantial amounts of ROS, and causes cell damage. Vitamin K2 participates in electron transfer in the oxidative respiratory chain and is an effective antioxidant. We used a DCFH-DA probe to detect the content of ROS in cells to explore the effect of vitamin K2 on cellular oxidative stress. Compared with the control group, intracellular DCFH-DA fluorescence was significantly increased in the 6-OHDA treatment group, indicating increased ROS accumulation. However, the ROS-dependent fluorescence signal was significantly reduced in the group treated with vitamin K2 (Figure 3). This shows that vitamin K2 can effectively remove the ROS generated by 6-OHDA and relieve cellular oxidative stress.

### 3.4. Vitamin K2 Inhibits Mitochondrial Depolarization Induced by 6-OHDA

A hallmark event in the early stages of apoptosis is a decrease in the mitochondrial membrane potential. The JC-1 probe can detect the mitochondrial membrane potential. When JC-1 forms aggregates that can produce red fluorescence, it means that the mitochondrial membrane potential is high; but when it forms a monomer that can produce green fluorescence, it means that the mitochondrial membrane potential is low. Therefore, the change in cell membrane potential can be easily detected by the transition of red and green fluorescence. When the cells were exposed to 6-OHDA, a significant enhancement in the green fluorescence signal was observed, while the red/green ratio decreased compared to the control group. However, after post-treatment with vitamin K2 for 6 h, the green fluorescence signal decreased, and the red/green ratio increased (Figure 4). This shows that vitamin K2 can restore the mitochondrial membrane potential and inhibit mitochondrial depolarization caused by 6-OHDA.

### 3.5. Effects of Vitamin K2 on Mitochondrial Fusion and Division in 6-OHDA-Mediated Injury Cells

Mitochondria are dynamic organelles. Mitochondrial fission and fusion together constitute mitochondrial dynamics. These two opposing processes work together to maintain the shape, size, number, and physiological function of the mitochondria [22]. We analyzed the changes of mitochondrial fusion and fission-related proteins by using Western blot and RT-qPCR to explore the effect of vitamin K2 on mitochondrial fusion and fission. Compared with the control group, the mRNA (Figure 5A) and protein (Figure 5B) expression levels of mitochondrial fission-related proteins (MFN1 and MFN2) in the 6-OHDA treatment group decreased, while the expression level of mitochondrial mitogen-related proteins (DRP1) increased. For the post-treatment with vitamin K2, the expression levels of MFN1 and MFN2 were increased, and the expression level of DRP1 was significantly decreased. However, the expression of DRP1 was increased compared with the normal group (Figure 5A,B). These results indicated that vitamin K2 treatment inhibited the abnormal mitochondrial division and fusion caused by 6-OHDA and promoted normal mitochondrial division, thus accelerating the clearance of damaged mitochondria. It has been demonstrated that vitamin K2 is able to maintain the homeostasis of the mitochondrial network, improve mitochondrial dysfunction, and prevent the expansion of mitochondrial damage.

### 3.6. Effects of Vitamin K2 on Mitophagy and Mitochondrial Biogenesis in 6-OHDA-Mediated Injury Cells

Mitophagy is a process that specifically targets and degrades damaged mitochondria. The timely removal of damaged mitochondria can effectively block cell death caused by damaged mitochondria [23]. Among them, the degradation products of amino acids, fatty acids, and other small-molecule compounds can be recycled by the body to provide a material basis for mitochondrial biogenesis. Western blotting and RT-qPCR were used to analyze changes in mitophagy and mitochondrial biogenesis-related proteins in each group. Compared with the control group, the expression levels of mitophagy-related proteins p62 and LC3A in the 6-OHDA-treated group increased, whereas the expression levels of PGC-1α, NRF1, and TFAM were decreased, and they are associated with mitochondrial biogenesis. In addition, post-treatment with vitamin K2 decreased the expression of mitophagy-related protein p62; increased the expression of LC3A; and increased the expression of PGC-1α, NRF1, and TFAM (Figure 6). This indicates that vitamin K2 promoted mitophagy and mitochondrial biogenesis in 6-OHDA-injured cells. These results suggest that vitamin K2 improves mitochondrial dysfunction and alleviates cell damage by regulating the mitochondrial quality-control system.

### 3.7. Vitamin K2 Regulates Mitochondrial Quality-Control System by Activating Pink1/Parkin Signaling Pathway

Attenuated mitophagy due to the silencing of PINK1 and Parkin proteins is a common trigger in PD [24]. To explore whether the PINK1/Parkin signaling pathway is involved in the mitophagy process regulated by vitamin K2, RT-qPCR and Western blotting were used to analyze the mRNA (Figure 7A) and protein (Figure 7B) expression levels of PINK1 and Parkin, respectively. The results showed that 6-OHDA slightly increased the expression of PINK1 and Parkin proteins, compared to that in the control group. Compared to the 6-OHDA treatment group, the expression levels of PINK1 and Parkin in the group treated with vitamin K2 increased. These results suggest that the PINK1/Parkin signaling pathway participates in the activation of mitophagy induced by vitamin K2.

## 4. Discussion

Although the pathological mechanism of PD is still not fully elucidated, it has been demonstrated that oxidative-stress-induced mitochondrial dysfunction and autophagy and apoptosis homeostasis can lead to the deterioration of dopaminergic neurones in the substantia nigra susceptible to oxidation [25]. Therefore, since mitochondrial protection by antioxidants may be a potential therapeutic strategy for PD treatment. To this end, this research investigated the influence of vitamin K2 on oxidative stress and mitochondrial function to elucidate its anti-PD mechanism. The results showed that vitamin K2 can promote the balance of mitochondrial dynamics in mitochondrial dysfunction induced by high oxidative stress and regulate the mitochondrial quality-control loop through the Pink1/Parkin signaling pathway, thereby alleviating mitochondrial dysfunction and cell damage.

To verify the ability of vitamin K2 to regulate oxidative stress and mitochondrial dysfunction, SH-SY5Y cells were exposed to 6-OHDA, and the optimal dose was determined based on an appropriate dose gradient in an in vitro model. This study found that 300 µM of 6-OHDA could significantly induce cell death, but post-treatment with 30 µM of vitamin K2 for 6 h effectively inhibited the cell death induced by 6-OHDA. Moreover, vitamin K2 also has a significant anti-apoptotic effect through the upregulation of Bcl-2 and downregulation of Bax expression, which play the role of the apoptosis main switch. A previous study showed that PC12 cells’ viability was significantly reduced when exposed to Aβ(1–42) peptide and H2O2. However, this inhibitory effect was greatly alleviated when pretreated with vitamin K2 [26]. In astroglioma C6 cells, vitamin K2 also helps cells resist Aβ-induced toxicity [27]. Moreover, vitamin K2 can reduce cytotoxicity of SH-SY5Y cells induced by rotenone [19] and regulate neuronal cell death by inhibiting endoplasmic reticulum stress and total tau protein expression [28]. This indicates that vitamin K2 exerts anti-apoptotic and neuroprotective effects in neurodegenerative diseases.

Mitochondria are the main sites for oxidative phosphorylation and ATP synthesis in cells; they are also the main source of ROS in cells [29], and the vast majority of ROS in cells is generated by the mitochondrial electron transport chain [30]. Under normal physiological conditions, as a cell-signaling molecule, ROS is controlled at a low level and plays a particularly significant role in the process of signal transduction, for instance, cell proliferation, differentiation, and apoptosis. However, oxidative stress occurs when ROS generation is increased and/or antioxidant levels or activity are decreased, and the balance between ROS removal and the repair of damaged complex molecules is disrupted, resulting in mitochondrial dysfunction. This affects cell division and growth, leading to cell dysfunction. Moreover, 6-OHDA, a catecholamine analogue, causes specific degeneration of substantia nigra neurones, and the neurotoxicity caused by 6-OHDA is partly due to ROS production [31]. Previous studies have established that 6-OHDA contributes to neuronal death by affecting mitochondrial function, including reducing ATP production, disturbing ROS metabolism, and inducing apoptosis [32]. This result also showed that 6-OHDA impaired mitochondrial redox function, reduced mitochondrial membrane potential, and increased ROS production. A reduction in mitochondrial redox activity and mitochondrial membrane potential may play a significant role in 6-OHDA-induced dopaminergic neurotoxicity. However, vitamin K2 can alleviate these toxic effects induced by 6-OHDA. These suggest that ameliorating mitochondrial dysfunction may be a better approach for slowing the progressive loss of dopaminergic neurones associated with PD.

Mitochondria maintain the dynamic balance of the mitochondrial network through continuous fission and fusion, which is necessary for maintaining the normal shape, distribution, and function of the mitochondria. A dynamic imbalance can lead to changes in the mitochondrial structure, causing dysfunction. Mitochondrial fission is mainly regulated by dynamin-related protein 1 (DRP1). The process of mitochondrial fission can fragment irreparable mitochondria and remove them from the cell to maintain the quality of mitochondria and protect the normal function of the mitochondrial network [33]. However, abnormal mitochondrial fission can cause the mitochondria to split into fragments, which, in turn, affects mitochondrial function. Mitochondrial fusion is mainly regulated by mitofusin protein 1/2 (MFN1/2), which plays a significant role in the repair of damaged mitochondria. Studies have shown that, when human aortic endothelial cells are damaged, the expression of Drp1 is upregulated, resulting in the destruction of the mitochondrial structure and an increase in ROS generation. When the expression of Drp1 is inhibited, vascular endothelial cell damage is significantly improved [34]. According to our Western blot results, after treatment with 6-OHDA, the expression of MFN1/2 and DRP1 was significantly decreased and increased, respectively. Post-treatment with vitamin K2 promoted and inhibited the expression of MFN1/2 and DRP1, respectively, but the expression of DRP1 was promoted compared with the control group. This result indicated that post-treatment with vitamin K2 not only inhibited the abnormal mitochondrial fission and fusion caused by 6-OHDA but also promoted the normal fission of mitochondria, accelerated the clearance of damaged mitochondria, maintained the dynamic balance of the mitochondrial network, and improved mitochondrial dysfunction.

Mitophagy is primarily the targeted elimination of excess or damaged mitochondria. Mitophagy is regulated by key proteins in the autophagy pathway, such as p62 and LC3. Moreover, p62 is often found in protein aggregation diseases. Autophagy can selectively aggregate damaged organelles and misfolded proteins [35]. LC3A is a member of the LC3 protein family. Under the action of phosphatidylethanolamine, LC3Ⅰ is lipidated to LC3Ⅱ, and the formation of LC3Ⅱ-positive autophagosome means the continuous formation of vesicles [36]. Therefore, the downregulation of p62 and upregulation of LC3A may reflect enhanced autophagic activity. In our study, we found that post-treatment with vitamin K2 downregulated the expression of p62 and upregulated the expression of LC3A in 6-OHDA-injured cells. This result indicated that post-treatment with vitamin K2 promoted mitophagy and alleviated the damage caused by 6-OHDA to cells. To explore the mechanism by which vitamin K2 activates mitophagy, we detected changes in the expression of PINK1 and Parkin proteins. Parkin can be recruited to damaged mitochondria and is crucial for mitochondrial dynamics and autophagy [37]. PINK1 is a protein associated with Parkinson’s disease. In normal mitochondria, the PINK1 protein exists in the mitochondrial outer membrane and can function as a sensor for damaged mitochondria. Its stability is necessary for Parkin to recruit damaged mitochondria and stimulate mitochondrial autophagy. Its stability is necessary for Parkin to recruit damaged mitochondria and stimulation of mitophagy [38]. Studies have found that vitamin K2 promotes mitophagy by increasing the expression of PINK1 and Parkin, indicating that vitamin K2 may recruit the autophagy molecules p62 and LC3 through the PINK1/Parkin signaling pathway, thus forming autophagosomes and promoting mitophagy.

Additionally, vitamin K2 regulates mitochondrial biogenesis. PGC-1α, a key regulator of mitochondrial transcription, triggers NRF1 expression in nuclear DNA, which, in turn, affects TFAM expression in mitochondrial DNA [39] and ultimately promotes the transcription, replication, and stabilization of mtDNA [40]. According to our results, the expression of PGC-1α, NRF1, and TFAM was significantly increased under vitamin K2 treatment compared with that in the 6-OHDA group, suggesting that post-treatment with vitamin K2 promoted mitochondrial biogenesis in 6-OHDA-injured cells.

In summary, vitamin K2 can inhibit the production of ROS and relieve oxidative stress in cells, and repair mitochondrial dysfunction by regulating the mitochondrial quality-control loop. Thus, it can play an important role in the prevention of PD. In addition, it may play a role in other diseases related to oxidative stress and mitochondrial function, such as non-alcoholic fatty liver disease (NAFLD) [41] and cancer [42] associated with oxidative stress and altered mitochondrial metabolism.

## 5. Conclusions

In recent years, with the in-depth study of mitochondrial structure and function, the progressive development of PD has been found to be closely related to mitochondria. Many factors can cause mitochondrial damage. If the damaged mitochondria in dopaminergic neurones cannot be removed in time, it will cause apoptosis and necrosis of dopaminergic neurones, leading to the occurrence of PD. This study found that vitamin K2, as an antioxidant, has a protective effect against PD. On the one hand, it interacts with bcl-2 family proteins, upregulates the Bcl-2 protein expression level, and downregulates the Bax protein expression level to play the role of the main switch of apoptosis, thereby alleviating mitochondrial oxidative stress and cell damage. On the other hand, it regulates mitochondrial fission, mitochondrial fusion, mitophagy, and mitochondrial biogenesis to maintain the mitochondrial quality-control loop, thereby alleviating mitochondrial dysfunction caused by 6-OHDA and maintaining mitochondrial homeostasis. Vitamin K2 maintains the balance of mitochondrial fission/fusion by downregulating the aberrant expression of DRP1 and upregulating the expression of MFN1/2 in 6-OHDA-injured cells. Simultaneously, the autophagy molecules p62 and LC3 are recruited through the PINK1/Parkin signaling pathway to form autophagosomes, promote mitophagy, and degrade damaged mitochondria. Finally, the resulting degradation products are reused to generate new mitochondria. Therefore, the normal operation of the mitochondrial quality-control loop was maintained and the cellular damage caused by mitochondrial dysfunction was alleviated (Figure 8).

## Figures and Tables

**Figure 1 nutrients-14-01504-f001:**
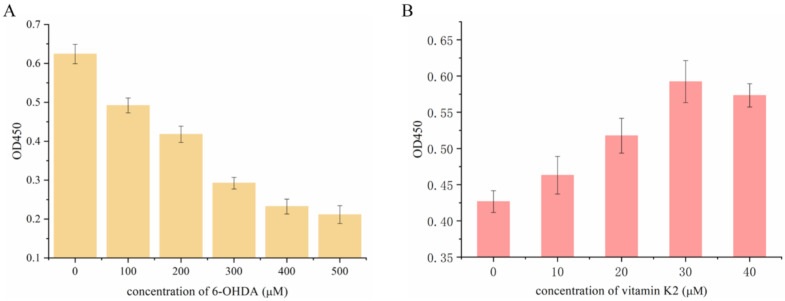
Measurement of SH-SY5Y cell viability after (**A**) treatment with different concentrations of 6-OHDA (0–500 µM) for 2 h and (**B**) 300 µM 6-OHDA treatment for 2 h and post-treatment with vitamin K2 for 6 h. The data are presented as mean ± SD (*n* = 3).

**Figure 2 nutrients-14-01504-f002:**
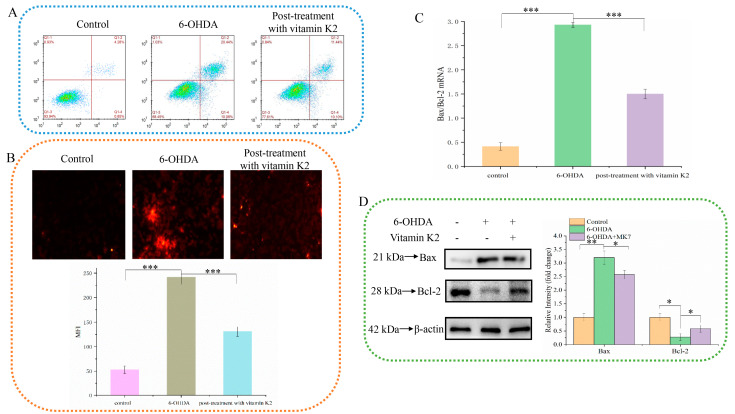
Effect of vitamin K2 on SH-SY5Y cell apoptosis in the presence of 6-OHDA. (**A**) Apoptotic cells were detected by flow cytometry. (**B**) Images of cells were obtained by fluorescence microscopy. All images were taken with a 20× objective. Images were taken in red modes, where red staining represents apoptosis cells, and representative graphs of the mean fluorescence intensity (MFI) from three different fields of view, using ImageJ software. (**C**) Measurement of Bax and Bcl-2 mRNA expression changes by RT-qPCR. (**D**) Changes in protein expression levels of Bax and Bcl-2 detected by Western blot; images were quantified by using the ImageJ software. The data are presented as mean ± SD (*n* = 3). Significant difference * *p* < 0.5, ** *p* < 0.05, and *** *p* < 0.01 vs. control.

**Figure 3 nutrients-14-01504-f003:**
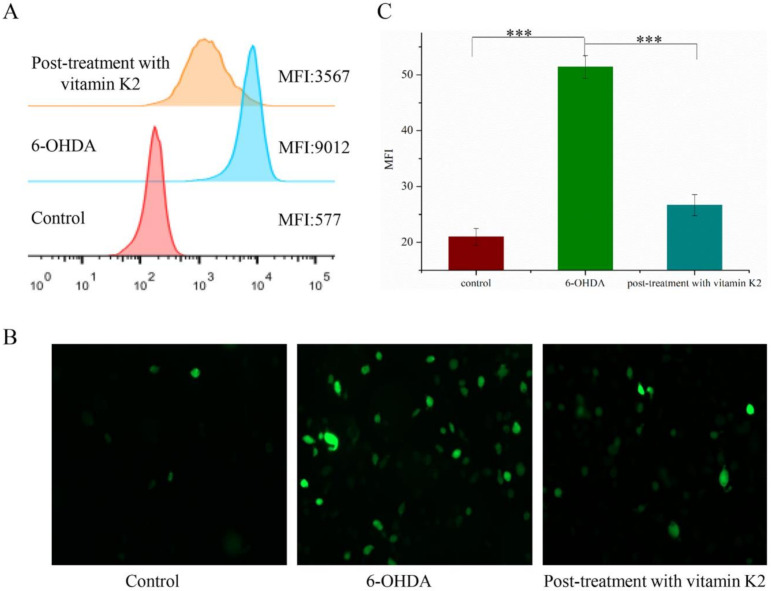
Analysis of reactive oxygen species level in SH-SY5Y cells. (**A**) Reactive oxygen species in cells were detected by flow cytometry. (**B**) Images of cells were taken by fluorescence microscopy. All images were taken with a 20× objective. Images were taken in green modes, where green staining represents ROS. (**C**) Representative graphs of the mean fluorescence intensity (MFI) from three different visual fields of view, using the ImageJ software. The data are presented as mean ± SD (*n* = 3). Significant difference *** *p* < 0.01 vs. control.

**Figure 4 nutrients-14-01504-f004:**
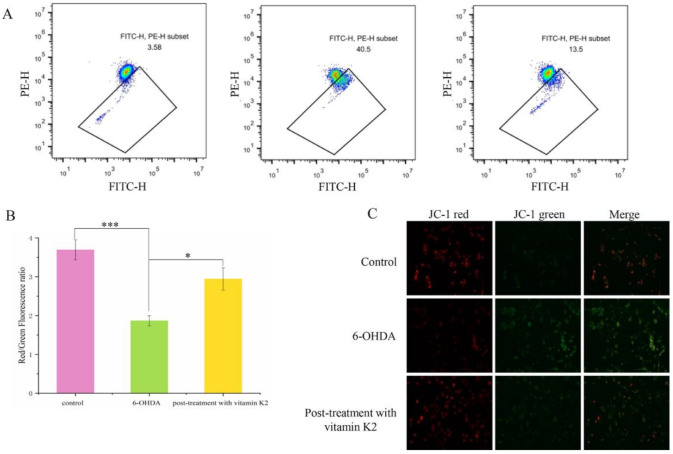
Measurement of the mitochondrial membrane depolarization state by using the JC-1 assay in SH-SY5Y cells. (**A**) Mitochondrial membrane potential in cells were detected by flow cytometry. (**B**) The red/green fluorescence ratio corresponding to the respective fluorescence images. (**C**) Images of cells were taken by fluorescence microscopy. All images were taken with a 20× objective. Images were taken in red and green modes, where green staining represents JC-1 monomers and red staining represents JC-1 aggregates. The data are presented as mean ± SD (*n* = 3). Significant difference * *p* < 0.5, and *** *p* < 0.01 vs. control.

**Figure 5 nutrients-14-01504-f005:**
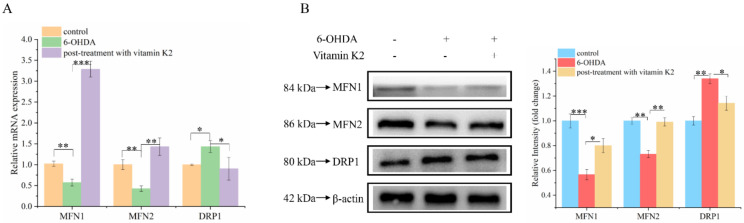
Effect of vitamin K2 on mitochondrial dynamic balance in 6-OHDA-injured cells. (**A**) Measurement of MFN1, MFN2, and DRP1 mRNA expression changes by using RT-qPCR. (**B**) Changes in protein expression levels of MFN1, MFN2, and DRP1 detected by Western blot; images were quantified by using the ImageJ software. The data are presented as mean ± SD (*n* = 3). Significant difference * *p* < 0.5, ** *p* < 0.05, and *** *p* < 0.01 vs. control.

**Figure 6 nutrients-14-01504-f006:**
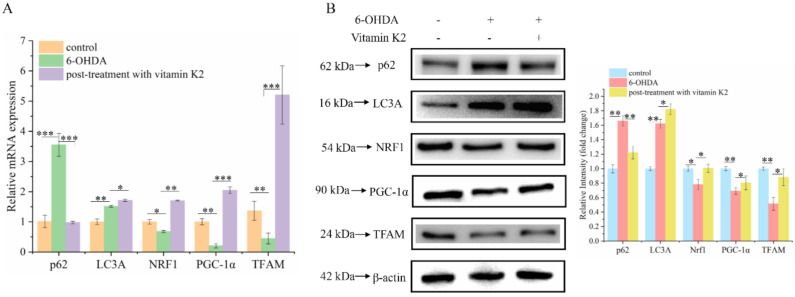
Effects of vitamin K2 on mitophagy and mitochondrial biogenesis in 6-OHDA-injured cells. (**A**) Measurement of p62, LC3A, PGC-1α, NRF1, and TFAM mRNA expression changes by RT-qPCR. (**B**) Changes in protein expression levels of p62, LC3A, PGC-1α, NRF1, and TFAM assessed by Western blot; images were quantified by using the ImageJ software. The data are presented as mean ± SD (*n* = 3). Significant difference * *p* < 0.5, ** *p* < 0.05, and *** *p* < 0.01 vs. control.

**Figure 7 nutrients-14-01504-f007:**
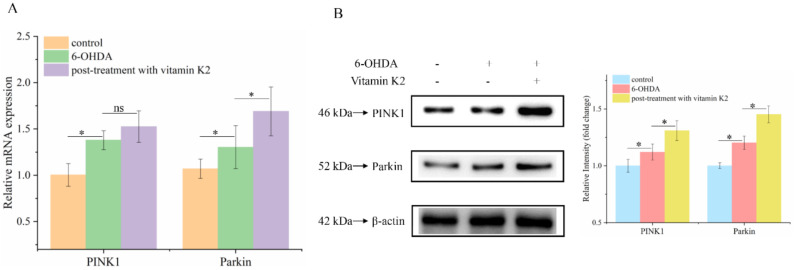
PINK1/Parkin signaling pathway was involved in the regulation of mitophagy by vitamin K2. (**A**) Measurement of PINK1 and Parkin mRNA expression changes by RT-qPCR. (**B**) Changes in protein expression levels of PINK1 and Parkin detected by Western blot; images were quantified by using the ImageJ software. The data are presented as mean ± SD (*n* = 3). Significant difference * *p* < 0.5 vs. control. (ns: no significance)

**Figure 8 nutrients-14-01504-f008:**
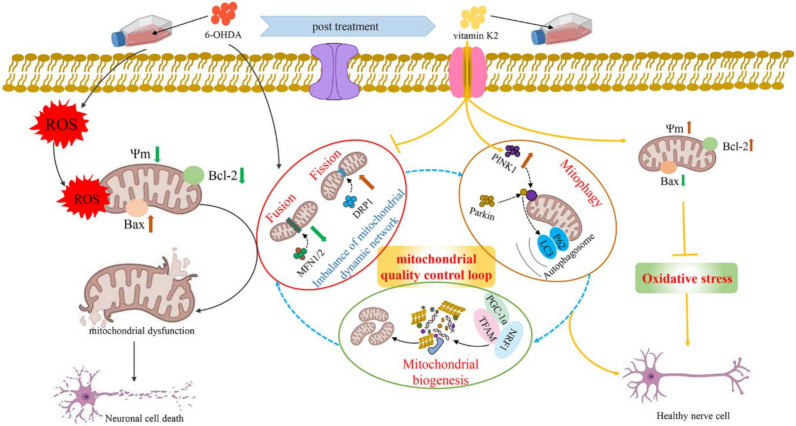
Schematic illustration of the protective role of vitamin K2 in 6-OHDA-induced cytotoxicity in SH-SY5Y cells. The 6-OHDA triggered ROS, causing oxidative stress in SH-SY5Y cells, which leads to mitochondrial depolarization (mitochondrial membrane potential reduction) and induction of mitochondria-mediated apoptosis, resulting in neuronal cell death. However, post-treatment with vitamin K2 maintains the balance of mitochondrial fission/fusion through the expression of regulatory proteins MFN1/2 and DRP1; promotes mitophagy by recruiting autophagy molecules p62 and LC3 through the PINK1/Parkin signaling pathway; and promotes mitochondrial biogenesis through the expression of regulatory proteins PGC-1α, NRF1, and TFAM. Thus, the normal operation of the mitochondrial quality-control loop (mitochondrial fusion, fission, autophagy, and biogenesis) is maintained, the intracellular oxidative stress is relieved, the cell membrane potential is restored, and the mitochondrial dysfunction is relieved. Moreover, by upregulating the expression of Bcl-2 protein and downregulating the expression of Bax protein to play the role of the main switch of apoptosis, the cell damage caused by mitochondrial dysfunction is reduced. Note: ROS, reactive oxygen species; 6-OHDA, 6-Hydroxydopamine; Ψm, mitochondrial membrane potential.

## Data Availability

Not applicable.

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
