# Peer review of "Vitamin K2 Modulates Mitochondrial Dysfunction Induced by 6-Hydroxydopamine in SH-SY5Y Cells via Mitochondrial Quality-Control Loop"

_nutrients, 2022, doi:10.3390/nu14071504_

Round 1

Reviewer 1 Report

The manuscript submitted by Hengfang Tang and coll., describes the activity of Vitamin K2 on modulating the mitochondrial dysfunction induced by 6-OH-Dopamine. This article is well written that could be published in present form. I have just a minor suggestion:

  • line 235 (page 6) after “… (Bcl-2)” : you should write “… (Figure 2C and D)” because that is not mentioned before.

Author Response

Thank you for your reminding, line 235 (page 6) has been modified according to your suggestions.

Reviewer 2 Report

I congratulate the efforts of the Authors for promoting new avenues in the discovery of potential targets for mitochondrial dysfunctions. Vitamin K2 is largely studied however its pleiotropic effects are inconclusively elucidated. The Authors found indications that Vitamin K2 is implicated in the regulation of the mitochondrial quality control loop, through the maintenance of the mitochondrial quality control system, thereby alleviating neuronal cell death mediated by mitochondrial damage.

  • The work is well-structured, the design sound and the methodology meticulously conducted. Statistics is appropriate, figures are fairly appreciable.
  • As a minor point, the theoretical framework and discussion might be enriched by comparing state-of-art and achieved results with other adjunct, tentative supplements to counteract similar dysfunctions. Please see: DOI: 10.1016/j.dld.2019.09.002

Author Response

Thank you for your reminding.  I supplemented the discussion by referring to the article "L-carnitine supplementation attenuates NAFLD progression and cardiac dysfunction in a mouse model fed with methionine and choline-deficient diet" (DOI: 10.1016/j.dld.2019.09.002), see revised manuscript lines 407-408 (page 11), lines 486-488 (page 12) and lines 489-491 (page 13).

This manuscript is a resubmission of an earlier submission. The following is a list of the peer review reports and author responses from that submission.